# Host JAK-STAT activity is a target of parasitoid wasp virulence strategies

**Susanna E. Brantley[1], Corinne M. Stouthamer[2], Pooja Kr[3], Mary L. Fischer[3], Joshua Hill[3], Todd A. Schlenke [2]\*, Nathan T. Mortimer [3,4]\***

**1** Department of Cell Biology, Duke University Medical Center, Durham, North Carolina, United States of America, **2** Department of Entomology, University of Arizona, Tucson, Arizona, United States of America, **3** School of Biological Sciences, Illinois State University, Normal, Illinois, United States of America, **4** Department of Biochemistry & Biophysics, Oregon State University, Corvallis, Oregon, United States of America

\* schlenke@email.arizona.edu (TAS); nathan.mortimer@oregonstate.edu (NTM)

**Data Availability Statement:** The data that support the findings of this study are publicly available from Open Science Framework with the identifier DOI 10.17605/OSF.IO/8SWHM.

## Abstract

Innate immune responses that allow hosts to survive infection depend on the action of multiple conserved signaling pathways. Pathogens and parasites in turn have evolved virulence factors to target these immune signaling pathways in an attempt to overcome host immunity. Consequently, the interactions between host immune molecules and pathogen virulence factors play an important role in determining the outcome of an infection. The immune responses of *Drosophila melanogaster* provide a valuable model to understand immune signaling and host-pathogen interactions. Flies are commonly infected by parasitoid wasps and mount a coordinated cellular immune response following infection. This response is characterized by the production of specialized blood cells called lamellocytes that form a tight capsule around wasp eggs in the host hemocoel. The conserved JAK-STAT signaling pathway has been implicated in lamellocyte proliferation and is required for successful encapsulation of wasp eggs. Here we show that activity of *Stat92E*, the *D. melanogaster* STAT ortholog, is induced in immune tissues following parasitoid infection. Virulent wasp species are able to suppress *Stat92E* activity during infection, suggesting they target JAK-STAT pathway activation as a virulence strategy. Furthermore, two wasp species (*Leptopilina guineaensis* and *Ganaspis xanthopoda*) suppress phenotypes associated with a gain-of-function mutation in *hopscotch*, the *D. melanogaster* JAK ortholog, indicating that they inhibit the activity of the core signaling components of the JAK-STAT pathway. Our data suggest that parasitoid wasp virulence factors block JAK-STAT signaling to overcome fly immune defenses.

## Author summary

Following infection, host immune responses are triggered to provide protection against the invading pathogen. The proper function of these responses depends on the activity of multiple immune signaling pathways. These pathways act in a coordinated manner to orchestrate the immune response, and any disruption can render the host susceptible to

**Funding:** Research reported in this manuscript was supported by National Institutes of Health (https://www.nih.gov/) Awards R35GM133760 to NTM and R01AI081879 to TAS. The funders did not play any role in the study design, data collection and analysis, decision to publish, or preparation of the manuscript.

infection. Because of this sensitivity, many pathogen species have evolved virulence mechanisms that target host signaling pathways and disrupt their function. We are using the interaction between *Drosophila melanogaster* and *Drosophila*-infecting parasitoid wasps to study this relationship between host signaling and pathogen virulence. Parasitoids infect fly larvae, and during infection, transfer virulence protein containing venom into the host, providing a mechanism to alter host signaling. This study is focused on one key immune signaling pathway, the highly conserved JAK-STAT pathway. We find that many virulent parasitoid species restrict the activity of the host JAK-STAT pathway, providing a clue as to their virulence strategy. JAK-STAT signaling plays many additional roles in health and disease, and so this system provides a good model to further understand this important pathway.

## Introduction

Innate immune mechanisms are evolutionarily ancient host defenses against invading pathogens. These innate mechanisms include both humoral and cell-mediated responses to infection and are coordinated by numerous immune signaling pathways [1]. Many of these pathways are highly conserved across taxa [2,3], including the signaling pathways underlying humoral responses such as the Toll/Toll-like Receptor pathway [4–7], and the pathways involved in cell-mediated immunity such as JAK-STAT, JNK and calcium signaling [8–13]. Following infection, the coordinated function of these signaling pathways is required for host resistance to infection [14,15]; the activation, duration and strength of each signaling event is therefore tightly regulated, and dysregulated immune signaling can result in immune deficiency [16–18]. In turn, pathogens have evolved an array of virulence factors that allow them to subvert host immune responses [19–21]. Conserved immune signaling pathways are commonly targeted by pathogen virulence factors across diverse host and pathogen taxa [12,22–26], reflecting their important role in host defense.

*Drosophila melanogaster* is a commonly used model of human health and disease, and studies into fly immunity have been key to understanding innate immune signaling [27–29]. Flies are naturally infected by a wide range of pathogens including microbes, viruses and macroparasites, and mount specific responses to each class of pathogen [30]. Parasitoid wasps are a common macroparasite of *Drosophila* species, and the *Drosophila*-parasitoid system has been used as a model to study the co-evolution and molecular mechanisms underlying host-parasite interactions [31–35]. Numerous parasitoid wasp species infect flies in nature, and during infection these wasps inject an egg along with virulence factor containing venom into the hemocoel of fly larvae. This triggers a host cellular immune response known as encapsulation [32,36]. In the cellular encapsulation response, circulating macrophage-like hemocytes known as plasmatocytes are activated by parasitoid infection and become the first hemocytes to encounter and bind to the foreign tissue [12,37]. Infection also triggers the production of lamellocytes, an induced hemocyte type that is required for the encapsulation of parasitoid eggs and other pathogens that are too large to be phagocytosed [38]. These specialized hemocytes can be derived from hemocyte precursors in the hematopoietic lymph gland as well as from the transdifferentiation of circulating and sessile plasmatocytes, via an intermediate pre-lamellocyte form (also known as podocytes or lamellocyte intermediates) [39–45]. Following their production, lamellocytes migrate towards and spread over the foreign tissue, forming a continuous capsule and killing the encapsulated object [37,46,47].

The production of lamellocytes is linked with activity of the JAK-STAT signal transduction pathway [48]. JAK-STAT pathway activation is initiated by the production and release of cytokines from the Unpaired (Upd) family: Upd1, Upd2, and Upd3 [49–52]. Upd proteins are secreted cytokines that bind to and activate the Domeless (Dome) cytokine receptor [50,53]. Activation of Dome triggers the activity of the Janus Kinase (JAK) ortholog Hopscotch (Hop), which in turn phosphorylates the Signal Transducer and Activator of Transcription (STAT) ortholog Stat92E [50,53,54]. Phosphorylated Stat92E can then enter the nucleus to drive transcription of downstream target genes [54–56]. JAK-STAT pathway activity leads to the production of lamellocytes both in response to macroparasite infection [48,57], and as a consequence of gain-of-function mutations in the *hop* or *Stat92E* genes, which cause the precocious differentiation of lamellocytes in naïve flies [58–61]. Conversely, loss-of-function mutations in *hop* lead to reduced hemocyte counts and weakened encapsulation ability [48,57].

Parasitoids that can successfully infect a host species are considered to be virulent with respect to that host, and this virulence is largely determined by parasitoid venom activity [62]. *Drosophila*-infecting parasitoid species have evolved diverse venom-mediated virulence strategies to overcome host immunity, and many of these mechanisms target host hemocyte production or function [12,26,46,63–70]. Notably, multiple species of *Drosophila*-infecting parasitoids employ venom factors that specifically target host lamellocytes [38,46,70–74], suggesting that inhibition of the production or activity of lamellocytes may be an effective parasitoid virulence strategy. Based on the key role of the pathway in lamellocyte production, we predict that parasitoids might suppress host lamellocyte production and encapsulation by specifically targeting the JAK-STAT pathway.

In this work, we ask if diverse *Drosophila*-infecting parasitoids species show evidence of suppressing JAK-STAT signaling, and whether these species target a common or distinct phase of JAK-STAT pathway activity. To test these ideas, we challenged larvae with a panel of parasitoid wasp species from the Family Figitidae. Many figitid wasps are virulent against *D. melanogaster* and have a wide range of venom activities that inhibit host lamellocyte function or target these immune cells for destruction. For instance, the well-studied parasitoid species *Leptopilina heterotoma* and *Leptopilina boulardi* both target host lamellocytes, but through the use of distinct strategies [63]. *L. heterotoma* venom induces lysis of host lamellocytes, resulting in a decreased population of this important cell type [75]. By contrast, *L. boulardi* venom contains a RhoGAP domain protein that induces morphological changes in lamellocytes and renders them incapable of participating in the encapsulation response in the absence of cell death [26]. Interestingly, previous gene expression data suggest that these species both may inhibit host JAK-STAT signaling by suppressing the expression of Upd ligands [63], adding support for the hypothesis that JAK-STAT signaling may be a common target of parasitoid venoms. Additional figitid species also target host lamellocytes. *Leptopilina victoriae*, a member of the *L. heterotoma* species group, inhibits the ability of lamellocytes to adhere to the encapsulated parasitoid egg by effecting the N-glycosylation state of lamellocyte surface proteins [46], however, it is unknown if the relevant venom activity inhibits the N-glycosylation process or acts as a deglycosylase. We also include a largely unstudied parasitoid species, *Leptopilina guineaensis*. The virulence strategy of *L. guineaensis* is unknown, but it is a virulent parasitoid of *D. melanogaster* [76].

There is additional evidence that figitid parasitoids can also indirectly disrupt host lamellocyte populations through the targeted elimination of hemocyte precursor cells. This has been previously reported for both *L. heterotoma* and *L. victoriae* [72]. In the case of *L. heterotoma*, precursor apoptosis happens in coordination with the lysis of mature lamellocytes [77], but the mechanism is less clear for *L. victoriae*. There are reported instances of intraspecies differences

in virulence [78], and it is possible that *L. victoriae* may provide an additional example of this phenomenon. While the *L. victoriae* strains used in this manuscript (LvHaw and LvPhil) inhibit lamellocyte function (via alterations to cell surface protein N-glycosylation) in the absence of decreased lamellocyte production [46], Chiu and Govind demonstrate that an unspecified *L. victoriae* strain induces apoptosis of hemocyte precursor cells in the hematopoietic lymph gland [72].

In addition, the figitid wasps contain two genera, *Leptopilina* and *Ganaspis*, which are found in overlapping regions of the world but belong to distinct species groups [79], allowing us to address the conservation of virulence strategies. Our study also includes *Ganaspis xanthopoda*, a virulent parasitoid of *D. melanogaster* [80]. *G. xanthopoda* infection depletes the host lamellocyte population similar to *L. heterotoma*, although the mechanism by which its venom acts, i.e., via lysis of mature lamellocytes, apoptosis of precursor cells, or another unidentified strategy, is unknown [81].

The Figitidae are particularly appropriate for this study since along with these virulent species, the family also includes species that are successfully encapsulated by *D. melanogaster*. These wasp species are referred to as avirulent, and can be assayed to investigate host immune function and for comparison with virulent species. In our study, we use two species which are avirulent with respect to *D. melanogaster*. *Leptopilina clavipes* is a specialist that successfully infects mushroom eating flies [82], and as such its venom is unable to inhibit *D. melanogaster* immunity [46]. *Leptopilina maia* is a newly discovered species with an unknown host range and virulence strategy [83], but given its avirulence with respect to *D. melanogaster*, it is likely that it also targets a distinct group of host species.

Our results show that this panel of virulent parasitoid wasp species suppress JAK-STAT pathway activity following infection. We find that parasitoids likely target the JAK-STAT pathway through distinct mechanisms, with different species inhibiting either the upstream activation or the activity of the core signaling components of the pathway. Our findings demonstrate that parasite virulence mechanisms have converged on a strategy of host JAK-STAT activity disruption.

## Results

### JAK-STAT pathway activation following parasitoid infection

To characterize the tissue specificity of JAK-STAT pathway activation in the context of the immune response to wasp infection, we assayed pathway activity using *Stat92E* activity reporter flies (*10XStat92E-GFP*) [84] in naïve larvae, and larvae infected with an avirulent parasitoid wasp. *10XStat92E-GFP* flies express GFP in response to JAK-STAT pathway activation (Fig 1A) such that GFP fluorescence can be used as a relative measure of pathway activity [84]. In naïve reporter flies, GFP expression was observed in cells of the gut (S1A Fig), but no expression was observed in immune tissues such as the hemocytes (Fig 1B and 1C), fat body (Fig 1F), body wall muscle (Fig 1G) or lymph gland (S1B Fig). To characterize reporter expression following infection, we challenged *10XStat92E-GFP* larvae with the avirulent parasitoid *Leptopilina clavipes* (strain LcNet), a species that induces a robust immune response and is usually encapsulated by *D. melanogaster* [46]. We found that LcNet infection alters the pattern of *10XStat92E-GFP* activity: at 48 hours post infection (hpi), GFP fluorescence was detected in hemocytes (Fig 1D and 1E), fat body (Fig 1H), and body wall muscles (Fig 1I), but not in the lymph gland (S1C Fig). In addition, as previously reported [48,57], we show that larvae heterozygous for a *Stat92E* mutant allele (*Stat92E^{06346},ry / +, ry*) have a significantly reduced ability to encapsulate LcNet eggs (F = 13.5, p = 0.021) (Fig 1J). Our data suggest that JAK-STAT

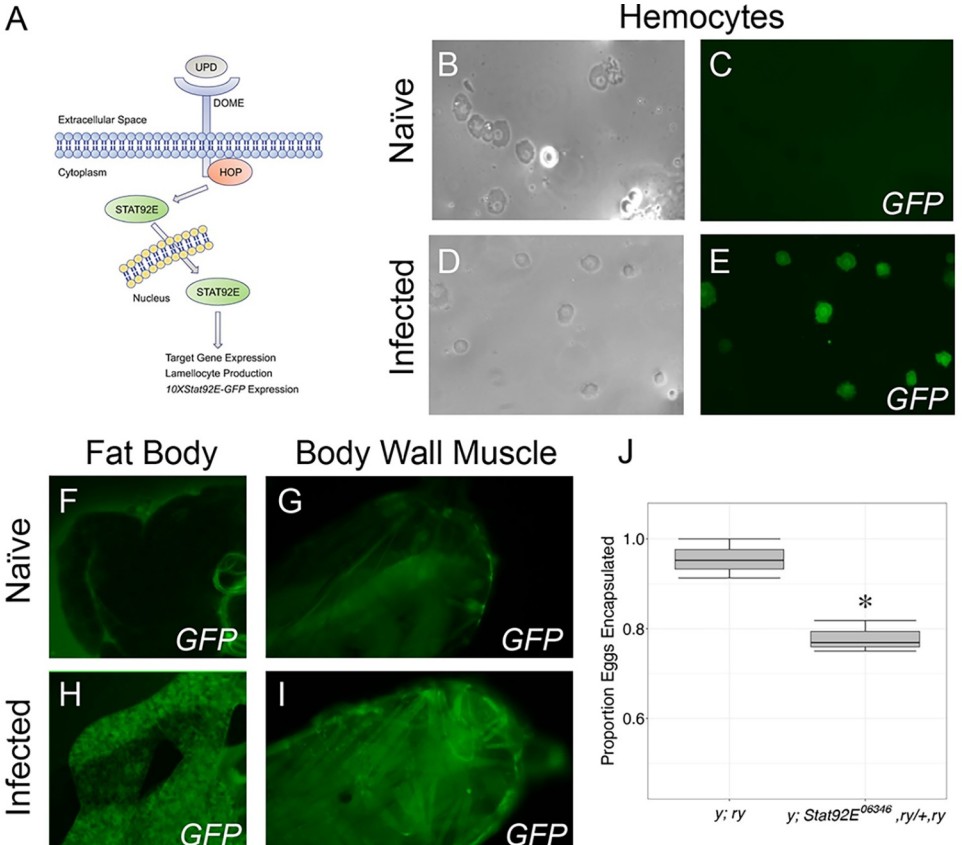

**Fig 1.** *Drosophila melanogaster* **JAK-STAT signaling pathway activity is induced in immune tissues following wasp infection.** (A) A schematic to illustrate the *D. melanogaster* JAK-STAT pathway. The pathway can be activated by the interaction between *upd* and *dome* or by gain of function mutations in *hop* or *Stat92E*. The *10XStat92E-GFP* transgene expression serves as a readout of pathway activity. (B-I) GFP expression in *10XStat92E-GFP* larval hemocytes (B,D: brightfield, C,E: GFP), fat body (F,H) and body wall muscle (G,I) was imaged 48 hours following infection by the avirulent wasp *L. clavipes* (strain LcNet) and in age-matched naïve larvae as indicated. *10XStat92E-GFP* activity is induced in hemocytes (compare C vs E), fat body (F vs H) and body wall muscle (G vs I) following LcNet infection. (J) Encapsulation of LcNet eggs in larvae heterozygous for *Stat92E*[06346] compared to the genetic background control. The data are shown as box plots with the mean proportions of each of three replicates visualized as open circles (*y; ry* n = 65; *y; Stat92E*[06346], *ry/+, ry* n = 80). * p < 0.05 compared to *y; ry* genetic background control by Analysis of Variance of Aligned Rank Transformed Data test.

signaling in these tissues may play an important role in the host encapsulation response to macroparasite infection.

## Lamellocyte production and changes to hemocyte concentration are induced by JAK-STAT activity in diverse tissues

*D. melanogaster* resistance to parasitoid infection is influenced by the total number of circulating hemocytes, the induction of lamellocytes, and also the concentrations of hemocyte types following infection [48,76,85–87]. To investigate the tissue specific roles of JAK-STAT activity in inducing changes in the host hemocyte population, we expressed a constitutively active form of *Stat92E* (*Stat92E*[CA]) [61] in immune tissues using the *UAS-GAL4* system [88]. Specifically, we expressed *Stat92E*[CA] in hemocytes using *He-Gal4*, in fat body using *C833-Gal4*, in body wall muscle using *Mef2-Gal4*, in the Posterior Signaling Center (PSC) of the lymph gland using *Dot-Gal4*, and in the Medullary zone (MZ) of the lymph gland using *unc-5*[GMR93E08]

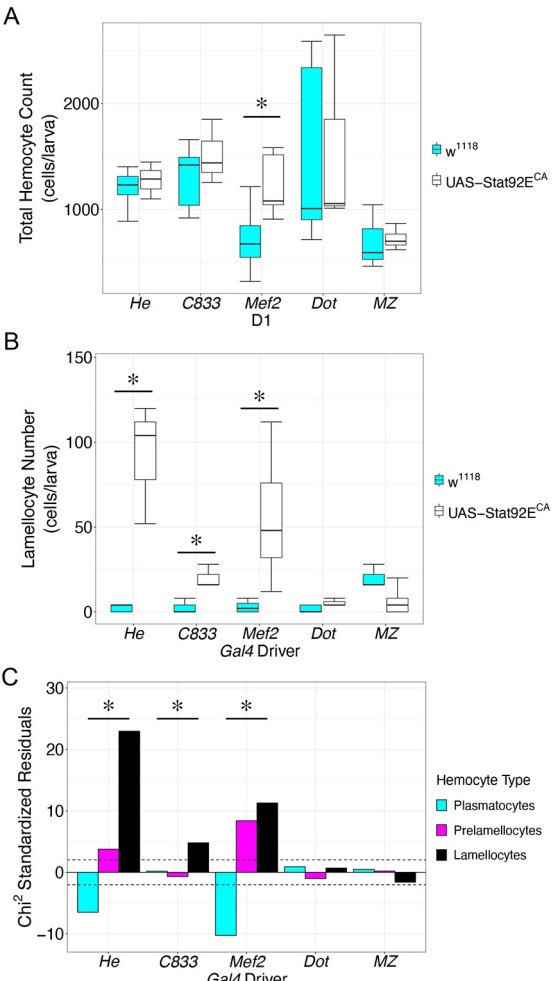

**Fig 2. JAK-STAT activity in immune tissues is sufficient for lamellocyte production.** (A) The total number of circulating hemocytes from larvae expressing a constitutively active form of *Stat92E* (*Stat92E^CA*, white) compared to a *w^1118* control cross (cyan). The data are shown as box plots with mean hemocyte number of each replicate visualized as open circles. (B) Lamellocyte numbers from larvae expressing a constitutively active form of *Stat92E* (*Stat92E^CA*, black) compared to *w^1118* controls (cyan). The data are shown as box plots with mean lamellocyte number of each replicate visualized as open circles. For A-B, * indicates p < 0.05 compared to the respective *w^1118* control cross by Welch two sample t-test. (C) χ² standardized residuals plotted for plasmatocytes (cyan), prelamellocytes (magenta) and lamellocytes (black) following expression of *Stat92E^CA* in each tissue. Hemocyte concentrations were significantly altered by *Stat92E^CA* expression in hemocytes, fat body and body wall muscle. * indicates p < 0.05 compared to the respective *w^1118* control cross by χ² test.

(hereafter referred to as *MZ-Gal4*) [60,89–92]. For each experiment we determined the number of each type of circulating hemocyte. The effect of *Gal4* expression was controlled for by outcrossing each *Gal4* line to the *w^1118* genetic background control.

We found that hemocyte-specific expression of *Stat92E^CA* failed to induce an increase in the total number of circulating hemocytes (Fig 2A), but that this expression led to a significant increase in the number of lamellocytes (Fig 2B). To further explore the effect of JAK-STAT signaling in hemocytes, we used the χ² test to determine whether the expression of *Stat92E^CA* in hemocytes altered the observed distribution of hemocytes across each type compared to the expected numbers calculated from the background control. We found that *Stat92E^CA* expression had a significant effect on the concentration of hemocyte types (χ² = 548.8, p = 4.99 x

$10^{-4}$). To investigate the effect on each hemocyte type we plotted the $\chi^2$ residuals for each cell type (Fig 2C), as a measure of the contribution of each cell type to the overall effect. This demonstrates that the concentration of plasmatocytes is decreased in these larvae coincident with an increase in the concentration of prelamellocytes and lamellocytes, suggesting that $Stat92E^{CA}$ expression in hemocytes is driving the transdifferentiation of plasmatocytes into lamellocytes via the prelamellocyte intermediate stage, as is also observed following parasitoid infection [39] or upon organism-wide JAK-STAT activation [45].

Further, we found that expression of $Stat92E^{CA}$ in the fat body, the major site of humoral immune signaling, promoted lamellocyte differentiation without significant changes to the total number of circulating hemocytes (Fig 2A and 2B). Fat body expression of $Stat92E^{CA}$ also altered the overall concentration of hemocytes ($\chi^2$ = 23.41, p = 1.99 x $10^{-3}$). In contrast to the case of $Stat92E^{CA}$ expression in hemocytes, this alteration in the fat body appeared to be driven solely by an increase in lamellocytes (Fig 2C). This suggests that fat body expression of $Stat92E^{CA}$ may induce the production of lamellocytes from non-circulating precursor cells. These contradictory results may be reconciled by considering that the number of lamellocytes produced is markedly less when $Stat92E^{CA}$ is expressed in fat body in comparison with hemocyte expression. Following fat body $Stat92E^{CA}$ expression, lamellocytes only account for 1.3% of the circulating hemocytes. This modest increase may not be sufficient to significantly impact the total number of hemocytes, and indeed we see that fat body $Stat92E^{CA}$ expression caused a small, but not statistically significant increase in hemocyte number (1516 ± 176 hemocytes/larva in $Stat92E^{CA}$ expression compared to 1313 ± 90.1 hemocytes/larva in control, p = 0.189). This suggests that the contribution of fat body signaling to lamellocyte production is limited, perhaps explaining why inhibition of JAK-STAT signaling in the fat body has minimal consequences for host defense following infection [57].

The larval body wall muscle has recently been shown to play a vital role in host defense to parasitoid infection, an effect mediated by JAK-STAT signaling [57]. Accordingly, we found that muscle-specific expression of $Stat92E^{CA}$ with the *Mef2-Gal4* driver was sufficient to induce both an increase in the total number of circulating hemocytes (Fig 2A) and the production of lamellocytes (Fig 2B), and altered the overall proportion of cell types in the hemocyte population ($\chi^2$ = 203.6, p = 5.0 x $10^{-4}$). The increase in total circulating hemocytes suggests that muscle JAK-STAT activity is likely triggering the production of hemocytes via the release from sessile hemocyte populations or the lymph gland into circulation. These newly produced hemocytes are likely differentiating into lamellocytes as reflected by the increased lamellocyte numbers and increases in the concentrations of both prelamellocytes and lamellocytes (Fig 2C).

Finally, we expressed $Stat92E^{CA}$ in two distinct cell populations within the lymph gland: the PSC and MZ. We found that $Stat92E^{CA}$ expression in either of these lymph gland cell types did not induce an increase in hemocyte number or lamellocyte production (Fig 2A and 2B), nor did it alter the concentration of hemocyte types in circulation (*Dot-Gal4*: $\chi^2$ = 1.467, p = 0.42; *MZ-Gal4*: $\chi^2$ = 2.549, p = 0.28) (Fig 2C). This is consistent with the described role of JAK-STAT signaling in restricting hematopoiesis within the lymph gland [93–95], and the lack of pathway activity in lymph gland cells following infection (S1C Fig).

## The virulence mechanisms of multiple parasitoid species target the JAK-STAT pathway

To test whether *Drosophila*-infecting parasitoid species can manipulate host JAK-STAT signaling, we challenged *Stat92E* activity reporter flies (*10XStat92E-GFP*) with a panel of diverse wasp species and strains (Fig 3). We found that immune challenge induced wasp-specific

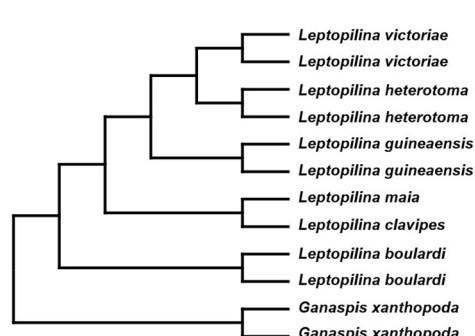

| Strain | Place Collected | Date Collected |
|---|---|---|
| LvPhil | Consolacion, Cebu, Phillippines | 2007 |
| LvHaw | Kaimuki, Oahu, HI, USA | 2009 |
| Lh14 | Winters, CA, USA | 2002 |
| LhSw | Uppsala, Sweden | 2007 |
| LgCam | Yaounde, Cameroon | 1998 |
| LgSA | False Bay, South Africa | 1999 |
| LmAtl | Atlanta, GA, USA | 2013 |
| LcNet | Heerenbergh, Netherlands | 2000 |
| Lb17 | Winters, CA, USA | 2002 |
| LbG486 | Brazzaville, Congo | 1978 |
| GxUg | Masindi, Uganda | 2010 |
| GxHaw | Kaimuki, Oahu, HI, USA | 2009 |

**Fig 3. Phylogenetic tree of parasitoid species and strains used in this study.** Phylogeny was constructed using COI sequences given in S1 Table. The place and date of collection is also given for each strain.

patterns of *10XStat92E-GFP* activity in hemocytes (Fig 4A and 4B), fat body (Table 1 and S2 Fig), and body wall muscle (Table 2 and S3 Fig) at 24hpi and 48hpi. Infection with two species of avirulent wasps, *Leptopilina clavipes* (strain LcNet) and *Leptopilina maia* (strain LmAtl) [83,96] both induce a high level of *10XStat92E-GFP* activity in hemocytes (Fig 4A and 4B), fat body (Table 1) and body wall muscle (Table 2).

In hemocytes, the levels of *10XStat92E-GFP* activity were highly induced at 24hpi (Fig 4A) following LmAtl and LcNet infection, and although they began to decrease, remained significantly elevated at 48hpi (Fig 4B). In contrast, infection with multiple strains of the virulent parasitoid species *Ganaspis xanthopoda*, *Leptopilina boulardi*, *Leptopilina guineaensis*, *Leptopilina victoriae* and *Leptopilina heterotoma* [33,46,74,76,81] induced lower levels of GFP fluorescence in hemocytes 24hpi (Fig 4A) and 48hpi (Fig 4B). GFP fluorescence levels in hemocytes from *10XStat92E-GFP* larvae infected with either of two strains of *L. guineaensis* (LgCam and LgSA) or *L. heterotoma* (Lh14 and LhSw) were indistinguishable from naïve controls at both time points, suggesting that the venoms of these wasps are either able to prevent the upstream activation of the JAK-STAT pathway or to inhibit the activity of the core signaling components following pathway activation in hemocytes. A moderate level of *10XStat92E-GFP* fluorescence was induced in hemocytes following infection by at least one strain of *G. xanthopoda* (GxHaw), *L. boulardi* (Lb17) or *L. victoriae* (LvHaw), but these levels were significantly lower than those induced by *L. clavipes* infection (GxHaw: t = -7.91, p < 1.0 x 10$^{-4}$; Lb17: t = -4.83, p < 1.0 x 10$^{-4}$; LvHaw: t = -4.83, p < 1.0 x 10$^{-4}$, all compared with LcNet at 48hpi). These data suggest that venoms from these virulent wasp strains are able to limit JAK-STAT pathway activity in host hemocytes to below the levels induced by infection with an avirulent species.

In the fat body (Table 1), infection by LgCam induced a comparable level of *10XStat92E-GFP* activity to infection by the avirulent strains LcNet or LmAtl. Interestingly, a second strain of *L. guineaensis* (LgSA) did not induce GFP fluorescence in the fat body, underlining the potential differences in virulence mechanisms between strains [34,78]. Infection by either of two strains each of *L. boulardi* (Lb17 and LbG486) and *L. victoriae* (LvPhil and LvHaw) led to a low level of GFP fluorescence in the fat body following infection, but infection by the other virulent parasitoid species did not induce fat body *10XStat92E-GFP* activity, suggesting that these wasps inhibit JAK-STAT activation in this tissue.

In the body wall muscle (Table 2), infection by *G. xanthopoda* strain GxUg induced a moderate level of *10XStat92E-GFP* activity. With the exception of GxUg, none of the other virulent parasitoid species, including the *G. xanthopoda* strain GxHaw, induced a visible level of GFP fluorescence in the body wall muscle following infection. In contrast, avirulent strains LcNet

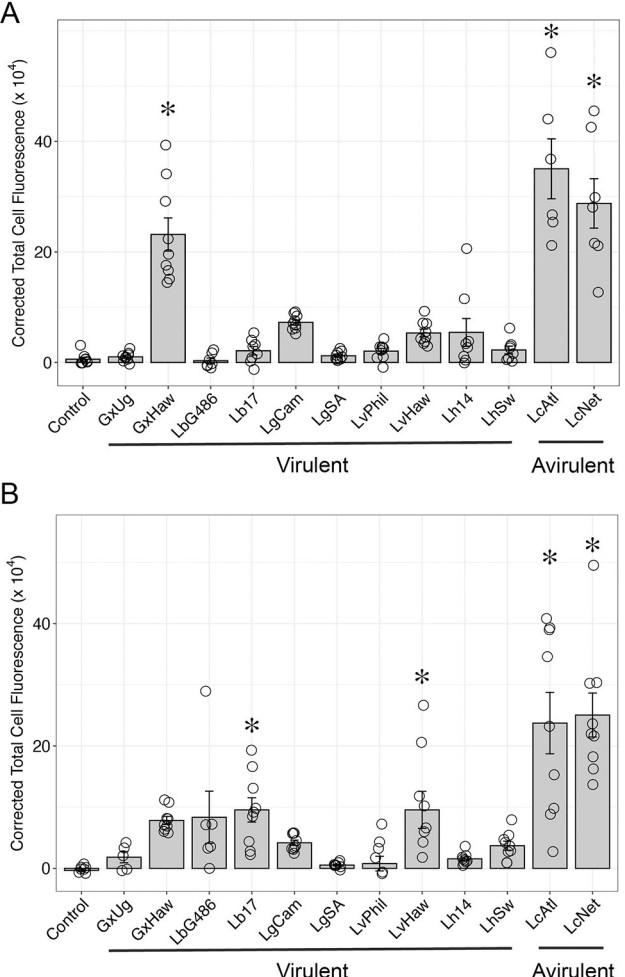

**Fig 4. GFP fluorescence is altered in *10XStat92E-GFP* hemocytes following wasp infection by multiple wasp strains.** GFP intensity measured as corrected total cell fluorescence in *10XStat92E-GFP* hemocytes at (A) 24 and (B) 48 hours post infection. Avirulent wasps LcNet and LmAtl induce the highest GFP fluorescence in comparison to multiple virulent wasp strains. Data are plotted as the average corrected total cell fluorescence ± standard error. The mean corrected total cell fluorescence of each replicate of 180 blood cells is visualized with open circles. * p < 0.05 compared to naïve control by Dunnett's test.

and LmAtl resulted in strong *10XStat92E-GFP* activity. Again, this suggests that the activation of the JAK-STAT pathway in body wall muscle is targeted by the venoms of these virulent parasitoid wasp species.

## Parasitoid virulence mechanisms target JAK-STAT at distinct steps of pathway activation

Our results show that JAK-STAT signaling is active in immune tissues following parasitoid infection and suggest that it can be modulated by wasp virulence mechanisms. Next, we wanted to test whether these parasitoid species are inhibiting the activation of the JAK-STAT pathway or the function of the core pathway components downstream of activation. To differentiate between these possibilities, we used the *hop* gain of function mutant fly strain, $hop^{Tum}$ [59]. $hop^{Tum}$ is a temperature-sensitive gain of function mutation, such that at restrictive temperatures *hop* activation becomes independent of upstream signaling leading to elevated

**Table 1. *10XStat92E-GFP* activity in fat body following wasp infection by multiple wasp strains.** (-, +, ++) indicates estimated observed fluorescence intensity. Examples of qualitative levels of GFP fluorescence are shown in S2 Fig.

| Treatment | 24h post-infection | 48h post-infection |
|---|---|---|
| Naïve | - | - |
| GxUg | - | - |
| GxHaw | - | - |
| LbG486 | + | + |
| Lb17 | + | + |
| LgCam | ++ | ++ |
| LgSA | - | - |
| LvPhil | - | + |
| LvHaw | - | + |
| Lh14 | - | - |
| LhSw | - | - |
| LmAtl | ++ | ++ |
| LcNet | ++ | ++ |

JAK-STAT signaling even in the absence of pathway induction by *upd* cytokines [58]. This results in the production of large numbers of lamellocytes in naïve flies [54]. Lamellocytes are adhesive cells, and in the absence of parasitoid eggs or another target for encapsulation in *hop^Tum* mutants, they bind to each other resulting in melanized cell nodules that are often referred to as "melanotic tumors" (Fig 5A) [97]. Because *hop^Tum* mutants express a constitutively active form of JAK, we predict that parasitoid species that inhibit pathway activation upstream of JAK would have no effect on the melanotic tumor phenotype, whereas parasitoid species that block the function of the core components of the pathway would suppress the melanotic tumor phenotype.

We infected *hop^Tum* flies with the panel of wasp species and strains and observed the effect of wasp infection on the penetrance of the melanotic tumor phenotype. Infection by multiple strains of two wasp species, *L. guineaensis* (LgCam and LgSA) and *G. xanthopoda* (GxUg and GxHaw) significantly reduced the penetrance of the melanotic tumor phenotype in *hop^Tum*

**Table 2. *10XStat92E-GFP* activity in body wall muscle tissue following wasp infection by multiple wasp strains.** (-, +, ++) indicates estimated observed fluorescence intensity. Examples of qualitative levels of GFP fluorescence are shown in S3 Fig.

| Treatment | 24h post-infection | 48h post-infection |
|---|---|---|
| Naïve | - | - |
| GxUg | + | + |
| GxHaw | - | - |
| LbG486 | - | - |
| Lb17 | - | - |
| LgCam | - | - |
| LgSA | - | - |
| LvPhil | - | - |
| LvHaw | - | - |
| Lh14 | - | - |
| LhSw | - | - |
| LmAtl | ++ | ++ |
| LcNet | ++ | ++ |

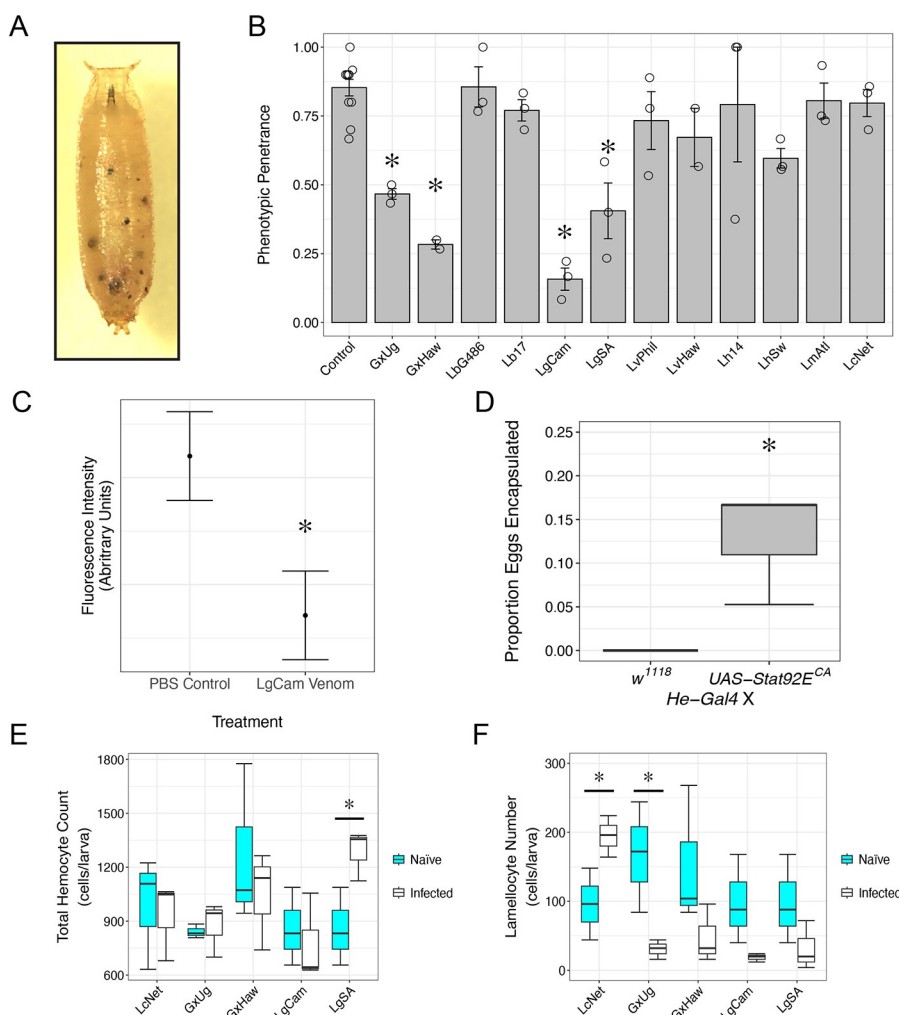

**Fig 5. Wasp virulence strategies target host encapsulation ability downstream of JAK.** (A) The $hop^{Tum}$ melanotic tumor phenotype observed in a mutant pupa. (B) Penetrance of $hop^{Tum}$ phenotype 72 hours after wasp infection and in age-matched naïve control. Data are plotted as the average penetrance ± standard error of replicates of thirty observed pupae. The mean penetrance of each replicate is visualized with open circles. * $p < 0.05$ compared to naïve control by Dunnett's test. (C) Calculated relative fluorescence intensity of GFP in primary hemocytes isolated from $hop^{Tum}$; *10XStat92E-GFP* larvae and incubated with LgCam venom or PBS control. Data are displayed as the mean fit (point) of the effect of incubation on fluorescence intensity ± standard error. * $p < 0.05$ compared to PBS control treated hemocytes by Analysis of Variance. (D) Encapsulation of LgCam eggs in larvae expressing $Stat92E^{CA}$ in hemocytes compared to $w^{1118}$ control cross. Data are expressed as mean proportion encapsulated ± standard error of three replicates. The mean encapsulation proportion of each replicate is visualized with open circles (*He-GAL4 x $w^{1118}$* n = 50; *He-GAL4 x UAS-Stat92E$^{CA}$* n = 55). * $p < 0.05$ compared to $w^{1118}$ control cross by Analysis of Variance of Aligned Rank Transformed Data test. (E-F) Total count of circulating hemocytes (E) and lamellocyte number (F) in $hop^{Tum}$ naïve larvae (cyan) or $hop^{Tum}$ larvae infected by the indicated parasitoid (black). The data are shown as box plots with mean hemocyte (E) or lamellocyte (F) number of each replicate visualized as open circles. * indicates $p < 0.05$ compared to naïve larvae by Welch two sample t-test.

flies (Fig 5B). These findings suggest that the virulence mechanisms of *L. guineaensis* and *G. xanthopoda* may act by inhibiting JAK-STAT pathway function downstream of activation.

We further explored this idea focusing on the virulent *L. guineaensis* strain LgCam. LgCam infection led to a significant decrease both in tumor penetrance in the $hop^{Tum}$ mutant background (Fig 5B), and in *10XStat92E-GFP* activity in hemocytes (Fig 4A and 4B) and body wall muscle (Table 2). To test the hypothesis that LgCam venom blocks JAK-STAT function

downstream of pathway activation, we incubated primary hemocyte cultures derived from $hop^{Tum}$; 10XStat92E-GFP larvae with purified LgCam venom in phosphate buffered saline (PBS). Hemocytes from these larvae have constitutively elevated JAK-STAT activity and 10XStat92E-GFP expression, and we predict that if LgCam venom is sufficient to block JAK-STAT pathway activity downstream of hop activation, then we will see decreased GFP fluorescence in hemocytes incubated with LgCam venom compared to hemocytes incubated with PBS control. We found that 1 hour of LgCam venom incubation was sufficient to significantly lower GFP fluorescence intensity compared to incubation with PBS alone (F = 13.2, p = 3.4 x $10^{-4}$) (Fig 5C). This suggests that LgCam venom components are inhibiting JAK-STAT via core pathway activity, rather than through an indirect effect of infection or pathway activation.

To test whether this JAK-STAT inhibition contributes to parasitoid success, we expressed $Stat92E^{CA}$ in hemocytes and measured the encapsulation rate of LgCam eggs. If LgCam virulence is dependent on inhibiting JAK-STAT pathway activity in hemocytes, we would expect that $Stat92E^{CA}$ expression would confer some degree of host resistance. We found that while none of the eggs in control larvae were encapsulated, flies expressing $Stat92E^{CA}$ in hemocytes showed a significantly increased rate of LgCam encapsulation (F = 36, p = 3.88 x $10^{-3}$) (Fig 5D). The virulence mechanism of L. guineaensis was previously unknown, but our findings support the idea that LgCam virulence targets core JAK-STAT pathway function, likely at the level of Stat92E activity.

## Suppression of the $hop^{Tum}$ melanotic tumor phenotype is correlated with alterations to the circulating hemocyte population

To determine how the virulence strategies of these virulent parasitoid species affect hemocyte production in $hop^{Tum}$ larvae, we measured the numbers of the different hemocyte types at 48hpi. In wild type flies infected by the avirulent parasitoid strain LcNet, increased numbers of lamellocytes are observed at 48hpi [46]. We found that LcNet infection also induced higher numbers of circulating lamellocytes in the $hop^{Tum}$ background compared to naïve $hop^{Tum}$ larvae (t = 2.85, p = 0.03) (Fig 5E and 5F), indicating that $hop^{Tum}$ mutants are capable of responding to parasitoid infection by further elevating lamellocyte numbers. We found that infection with strains of L. guineaensis and G. xanthopoda did not lead to a decrease in the total number of circulating hemocytes (Fig 5E). However, we found that infection by the G. xanthopoda strain GxUg led to a significant decrease in lamellocyte numbers in infected $hop^{Tum}$ mutant larvae (t = -2.89, p = 0.047) (Fig 5F). Infection by the closely related strain GxHaw and the L. guineaensis strains LgCam and LgSA also led to a consistent decrease in lamellocyte numbers (Fig 5F), although these values were not below an α value of 0.05 (GxHaw: t = -1.65, p = 0.104; LgCam: t = -2.133, p = 0.082; LgSA: t = -1.56, p = 0.106).

Based on previous findings that the concentration of hemocyte types in circulation is an important factor in host defense [48] and our observation that ectopic JAK-STAT expression in immune tissues altered this concentration (Fig 2C), we hypothesized that there may be a correlation between melanotic tumor penetrance and the concentration of hemocyte types. We determined the proportion of each hemocyte type in $hop^{Tum}$ larvae infected by all parasitoid species and strains and tested for a correlation between these proportions and the penetrance of the melanotic tumor phenotype following the respective infection. We found that relative numbers of plasmatocytes (ρ = -0.033, p = 0.915) (Fig 6A) or prelamellocytes (ρ = -0.341, p = 0.255) (Fig 6B) were not correlated with $hop^{Tum}$ melanotic tumor penetrance. However, we found a significant correlation between relative lamellocyte numbers and melanotic tumor phenotype penetrance (ρ = 0.555, p = 0.025) (Fig 6C), in which high tumor penetrance was associated with a higher proportion of circulating lamellocytes.

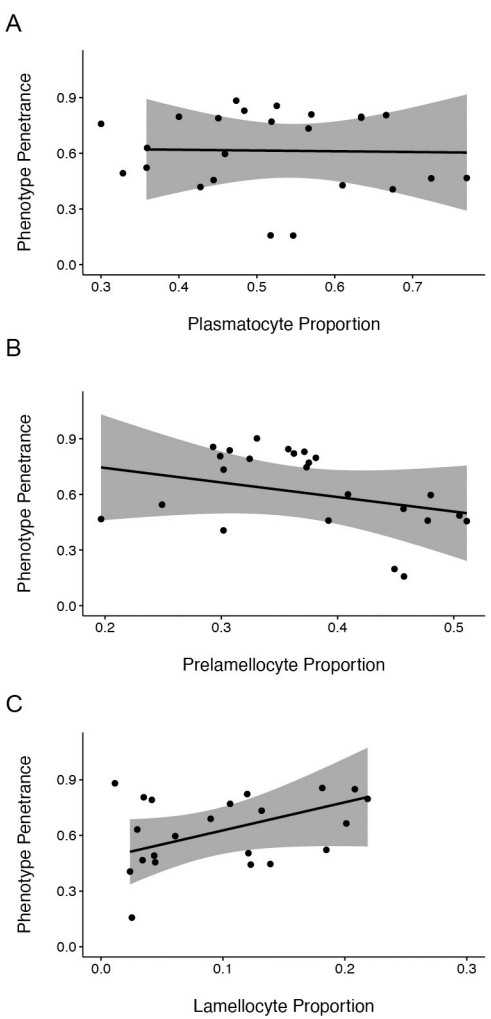

**Fig 6. Correlations between relative hemocyte counts and tumor penetrance in wasp-infected flies.** Scatterplots of relative counts of (A) plasmatocytes, (B) prelamellocytes, and (C) lamellocytes quantified 48hpi plotted against the penetrance of the $hop^{Tum}$ melanotic tumor phenotype following infection or in age-matched naïve larvae. Data are shown for $hop^{Tum}$ mutants infected by each wasp strain and naïve controls. The linear regression line for each correlation was calculated and plotted. 95% confidence intervals are shaded in gray. Plasmatocyte and prelamellocyte proportions do not correlate with percent tumor penetrance. In contrast, relative lamellocyte numbers are positively correlated with percent tumor penetrance.

## Discussion

Our data demonstrate that the JAK-STAT signaling pathway is active in *D. melanogaster* immune tissues following parasitoid wasp infection and that pathway activity in these tissues can induce the production of lamellocytes, likely via distinct mechanisms. JAK-STAT activation in hemocytes, via the hemocyte-specific expression of the constitutively active $Stat92E^{CA}$, led to the production of lamellocytes without an increase in the overall circulating hemocyte count. Previous studies that have shown through lineage tracing experiments that a subset of plasmatocytes transdifferentiate into lamellocytes as part of the immune response against parasitic wasps [39,40,45]. Our data suggest that this transdifferentiation may be stimulated by JAK-STAT activity specifically in hemocytes.

In agreement with recent work [57], we found that expression of $Stat92E^{CA}$ in the body wall muscle using the *Mef2-Gal4* driver induced an overall increase in the number of circulating hemocytes and the production of lamellocytes, suggesting that JAK-STAT activity in the body wall muscle may act through a non-autonomous signaling mechanism to induce the production of new hemocytes that differentiate into lamellocytes. We observe a similar trend when $Stat92E^{CA}$ expression is driven in the fat body by *C833-Gal4*, although far fewer lamellocytes were produced and the increase in hemocyte numbers did not reach significance. The origin of the *de novo* lamellocyte population following JAK-STAT activation in the body wall muscle or fat body is unclear from these data, but two hypotheses can be suggested. First, lamellocytes can be produced in the lymph gland from prohemocytes [98]. Perhaps a cytokine signal downstream of JAK-STAT activation is secreted from peripheral tissues to stimulate lamellocyte differentiation and release of new hemocytes into circulation from the lymph gland. A second hypothesis is that a cytokine produced by JAK-STAT signaling in the body wall muscle or fat body is received by the sessile hemocyte population. These blood cells are found adjacent to the body wall muscle and remain out of circulation in wild type, naïve flies. When marked with GFP, these cells can be observed as a distinct banding pattern on the fly cuticle [99,100]. However, upon wasp infection, this banding pattern disappears, suggesting that the sessile hemocytes have entered circulation [42,60]. Differentiation of these cells into lamellocytes may be the source of the excess lamellocytes we observed following expression of $Stat92E^{CA}$ in the body wall muscle or fat body.

Consistent with the induction of lamellocyte differentiation by $Stat92E^{CA}$ expression, we found that wasp infection led to JAK-STAT activity in the hemocytes, body wall muscle, and fat body. The requirement for JAK-STAT activity in the encapsulation response [48,57] suggests that the pathway could be a likely target for wasp virulence factors, and indeed we found that the virulent wasp strains we tested seemed to induce lower levels of JAK-STAT activity in the immune tissues of their fly hosts than the avirulent wasp strains LcNet and LmAtl. Based on our observations, we conclude that this pathway inhibition may take on several forms. For instance, in hemocytes, strains of *L. guineaensis* and *L. heterotoma* suppress JAK-STAT signaling to the level of naïve flies, suggesting a complete inhibition of pathway activation or function. On the other hand, hemocytes infected by GxHaw induced a peak of signaling at 24hpi but were indistinguishable from naïve flies by 48hpi, suggesting that GxHaw infection may act to shorten the duration of pathway activity. Conversely, JAK-STAT signaling in hemocytes of Lb17 and LvHaw infected flies was not active at 24hpi but showed elevated signaling at 48hpi. This is potentially indicative of a delay in pathway activation. Similar effects were seen in the ability of virulent wasps to manipulate host JAK-STAT activity in body wall muscle and fat body, although such patterns are more difficult to discern from qualitative data. The virulence of these wasp species suggests that along with the strength of pathway activity, the timing and duration of JAK-STAT signaling may also play a key role in host resistance.

The functional significance of the ability of wasps to target the JAK-STAT pathway as a virulence mechanism is evident in our observation that flies with overactive JAK-STAT pathway activity in hemocytes are less susceptible to the virulent wasp LgCam. The resistance conferred by $Stat92E^{CA}$ expression was incomplete, demonstrating that either LgCam is so effective at inhibiting the JAK-STAT pathway that its virulence mechanism can even perturb heightened pathway activity or that hemocyte-specific inhibition of JAK-STAT activity is not the sole mechanism of LgCam virulence. Instead, the inhibition of JAK-STAT signaling in the body wall muscle or the manipulation of distinct signaling pathways may also play important roles.

Our findings suggest that successful wasp parasites may be able to suppress host JAK-STAT signaling as a virulence strategy, either via preventing pathway activation by blocking upstream signaling mechanisms, or by directly inhibiting the core components of the

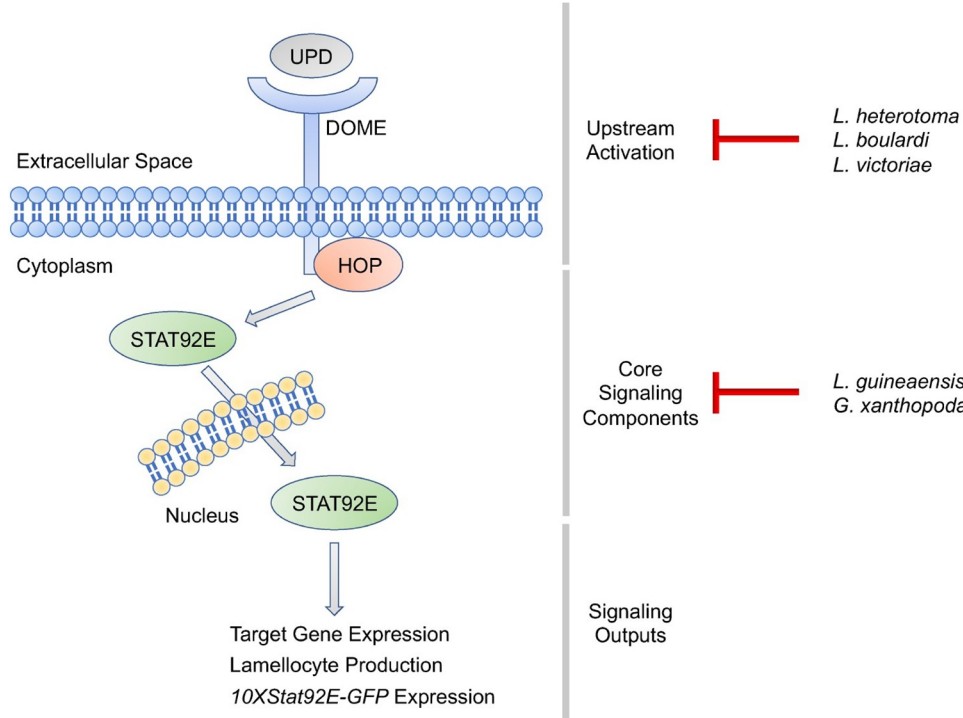

**Fig 7. *Drosophila*-infecting parasitoids inhibit the host JAK-STAT pathway.** Schematic illustrating the phases of JAK-STAT signal transduction: upstream activation, core signaling components, and signaling outputs including *10XStat92E-GFP* expression, and the phase of pathway activity inhibited by each parasitoid species we screened. For each species, the different strains appear to act at the same phase, so only species names are given for simplicity.

JAK-STAT signaling pathway (Fig 7). To distinguish between these possibilities, we tested the ability of the parasitoid strains to inhibit the melanotic tumor phenotype resulting from the $hop^{Tum}$ mutation. $hop^{Tum}$ is a gain of function mutation in the *D. melanogaster* JAK that renders its activation independent of upstream signaling [58,59]. We predicted that parasitoid strains that block the upstream activation of the JAK-STAT pathway would have no effect on the $hop^{Tum}$ mutant phenotype, whereas strains that directly inhibit core JAK-STAT signaling components would suppress melanotic tumor formation. We found that strains of two species of virulent wasps, *L. guineaensis* and *G. xanthopoda*, not only limited JAK-STAT activity in hemocytes and body wall muscle, but also decreased the penetrance of the tumor phenotype in $hop^{Tum}$ mutants. This suggests that *L. guineaensis* and *G. xanthopoda* virulence strategies likely target core components of the JAK-STAT pathway (Fig 7). This idea is supported by the finding that purified LgCam venom can inhibit $hop^{Tum}$ activity in primary hemocyte cultures, in the absence of immune induction or extrinsic pathway activation (Fig 5C). This finding also provides our first insight into the previously unknown virulence mechanism of *L. guineaensis*.

Strains of the additional virulent parasitoid species *L. heterotoma*, *L. boulardi*, and *L. victoriae* limited JAK-STAT pathway activity in host immune tissues but had no effect on the penetrance of the $hop^{Tum}$ phenotype. This suggests that these parasitoid strains may inhibit the upstream mechanisms leading to JAK-STAT activation (Fig 7). In *D. melanogaster*, the JAK-STAT pathway is activated when the Dome receptor binds to ligands from the Upd cytokine family [49–53]. In particular, Upd2 and Upd3 are linked to antiparasitoid immunity [57]. These cytokines are produced by multiple tissues following infection, injury, or other stress [57,101,102]. The virulence strategies of *L. heterotoma*, *L. boulardi*, and *L. victoriae* may

therefore block pathway activation by impairing the expression or function of the Upd proteins or by interfering with the ability of Dome to bind its ligands or transduce the cytokine signal by activating Hop. Previous studies of host gene expression found that there was no induction of expression of the *upd* genes following infection with either *L. heterotoma* strain Lh14 and *L. boulardi* strain Lb17 [63]. These cytokines are required for JAK-STAT pathway activation and the encapsulation of parasitoid eggs following infection [57], so the lack of *upd* induction suggests that this is a possible mechanism for the JAK-STAT inhibition seen in *L. heterotoma* and *L. boulardi*. *L. victoriae* may use a similar mechanism of upstream inhibition, although this remains to be thoroughly tested.

It has been demonstrated that individual wasp strains employ very specific virulence strategies against *D. melanogaster* [31,34,78], and our findings suggest that this extends to their anti-JAK-STAT strategies as well. These mechanisms of interference should be investigated further to better understand the ways that natural parasites may manipulate conserved signaling pathways involved in host immune responses. *D. melanogaster* is a widely used system for understanding JAK-STAT pathway functions. While there are four JAKs and seven STATs in the human system, *D. melanogaster* has only one JAK (*hop*) and one STAT (*Stat92E*), providing a much simpler system, but one with similar functionality and sequence conservation to the human JAK-STAT pathway [8,103]. Since its discovery more than twenty years ago, the JAK-STAT pathway has been shown to play an essential role in mammalian hematopoiesis similar to the mechanisms by which JAK-STAT signaling regulates cellular differentiation in *D. melanogaster*. Notably, the JAK-STAT signaling pathway is conserved across Drosophilids but might be activated in different tissues, times, and signal strengths across species. Experiments conducted in this study focused on a specific strain of *D. melanogaster*, but many of the parasitoid wasps' natural host ranges include other fly species, and there is also intraspecific variation in host immunity. In future work, we believe understanding the evolution of signaling and virulence mechanisms across strains and species could further our insights into the mechanisms of JAK-STAT signaling in the immune system. Drosophilids and *Drosophila*-infecting parasitoid wasps thus provide an excellent model system to study the highly conserved JAK-STAT signaling pathway *in vivo*.

## Materials and methods

### Insects

Flies were maintained on standard Drosophila medium of cornmeal, yeast, and molasses. The *He-Gal4* (BDSC_8699), *C833-Gal4* (BDSC_6988), *Dot-Gal4* (BDSC_6903), *unc-5^{GMR93E08}* (*MZ-Gal4*; BDSC_48001), *w^{1118}* (BDSC_5905), *y^1; ry^{506}* (BDSC_4405), *ry^{506},P{ry[+t7.2] = PZ} Stat92E^{06346}/TM3,ry^{RK},Sb^1,Ser^1* (BDSC_11681), and *hop^{Tum}* (BDSC_8492) strains were obtained from the Bloomington Drosophila Stock Center (Bloomington, IN) (RRID: SCR_006457). *Mef2-Gal4* (BDSC_27390) was provided by Dr. A. Vrailas-Mortimer [104]. The constitutively active *Stat92E* transgene *w^{1118};UAS-Stat92E^{ΔNΔC}* (referred to as *UAS-Stat92E^{CA}*) [61] and the JAK-STAT reporter strain *10xStat92E-GFP* (BDSC_26197) [84] were provided by Dr. E. Bach. Wasp strains (Fig 3) were obtained and maintained as described in [76]. Briefly, all wasps were maintained on the *D. melanogaster* strain Canton S, with the exception of avirulent strains LcNet and LmAtl, which were maintained on *D. virilis*. Strain LbG486 was provided by Dr. D. Hultmark, LcNet was provided by Dr J. van Alphen, GxUg, LgCam and LgSA were provided by Dr. J. Pool. All other wasp strains were collected by the Schlenke lab.

### Hemocyte counts

All crosses and experiments were carried out at 25°C with the exception of experiments carried out with *hop^{Tum}* temperature sensitive mutants, which were incubated at 28°C for up to 72

hours after a 24-hour wasp infection at 25˚C. For $Stat92E^{CA}$ expression hemocyte counts, crosses between *Gal4* virgin females and *UAS-Stat92E^{CA}* or $w^{1118}$ (wild type control) males were done at 25˚C, in three replicates per treatment. Wandering third instar larvae were selected for hemocyte counts. For $hop^{Tum}$ hemocyte counts, flies were allowed to lay eggs for 24 hours at 25˚C. Egg lay plates were then placed at 28˚C and incubated for 48 hours. Thirty early third instar larvae were picked from egg lay plates and transferred to 35mm diameter Petri dishes, in three replicates per treatment. For infection treatments, three female wasps were allowed to oviposit for 24 hours at 25˚C. Following wasp infection, wasps were removed and the infected and control naïve larvae were shifted to 28˚C. Hemocytes were counted at 24 and 48 hours after being moved to 28˚C. From each cross, three sets of 5 larvae were bled into 20 μL of 1X Phosphate Buffered Saline (PBS), pH 7.4 with 0.01% PTU (to prevent blood cell lysis and melanization). The hemolymph dilution was transferred to a disposable hemocytometer (Incyto C-Chip DHC-N01). Hemocytes from sixteen 0.25 x 0.25 x 0.1 mm squares were counted for each replicate.

## Imaging

Wasp infections of *10xStat92E-GFP* reporter flies were carried out as described above at 25˚C, and imaging took place 24 and 48 hours after wasp infection. Five larvae were dissected under immersion oil from each of three replicate plates. For quantification of *10xStat92E-GFP* corrected total cell fluorescence, 20 blood cells from each of the 5 larvae were analyzed for fluorescence in comparison to background using ImageJ software [12]. $hop^{Tum}$ flies were imaged as pupae after 72 hours of incubation at 28˚C following wasp infection to allow adequate time for development of the tumor phenotype. Penetrance was determined by phenotyping larvae for presence or absence of melanotic tumors.

## Wasp encapsulation

To measure wasp egg encapsulation ability, 30 late second instar larvae were placed on 35mm Petri dishes filled with standard Drosophila medium together with 3 female wasps at 25˚C. Larvae were dissected 72hpi and the number of encapsulated eggs and hatched wasp larvae were counted. Experiments were performed in triplicate.

## Venom purification

To separate the effects of venom activity from the infection and immune response process, venom was purified from dissected venom apparatuses as described in [105]. Briefly, dissected venom apparatuses were homogenized on ice in non-lysing conditions in 1X PBS, pH 7.4, supplemented with 0.5 mM EDTA and 1X Halt protease inhibitor cocktail (Thermo Scientific). Venom apparatus cells were pelleted by centrifugation at 12000 *g* at 4˚C, and the supernatant containing venom was used to assess activity.

## Ex vivo venom assays

To assess the ability of parasitoid venom to inhibit host JAK-STAT signaling, $hop^{Tum};10xStat92E-GFP$ larvae were raised at 28˚C and dissected to establish primary hemocyte cultures. Third instar larvae were dissected into 1X PBS, pH7.4, with 0.01% PTU in multi well slides (Tekdon). Primary hemocyte cultures were mixed with 10 wasp equivalents of venom or PBS control and transferred into Tali Cellular Analysis slides (Invitrogen). After a 1-hour incubation period, GFP fluorescence intensity was determined using a Tali Image-Based Cytometer (Invitrogen) as described in [70].

## Statistics

Welch two sample t-tests were used to compare immune cell count data between *UAS-Stat92E$^{CA}$* and *w$^{1118}$* crossed to each *Gal4* driver. To look for changes in the distribution of immune cell types, we used the $\chi^2$ test. For each *Gal4* driver, the proportion of each hemocyte type in the control *w$^{1118}$* cross used as the probability estimate and p values were simulated using 2000 replicates [106]. $\chi^2$ standardized residuals were plotted for each cell type in each comparison to estimate the contribution of cell type to the overall $\chi^2$ value. Residuals > |2| suggest a significant impact on the score [107].

Corrected total cell fluorescence values were analyzed by Analysis of Variance (ANOVA) followed by Dunnett's test for pairwise comparisons to the naïve control or LcNet infected larvae as indicated in the text. To analyze *hop$^{Tum}$* phenotypic penetrance data, we used generalized linear models with quasibinomial errors. Dunnett's test was used for pairwise comparisons to the control genotype. The proportion of encapsulated parasitoid eggs was compared between genotypes using the Analysis of Variance of Aligned Rank Transformed Data test. Correlation coefficients ($\rho$) between hemocyte concentrations and the penetrance of the *hop$^{Tum}$* melanotic tumor phenotype were calculated using Spearman's rank correlation. Mixed linear models, with replicate as a random effect, were used to test for differences in *10xStat92E-GFP* fluorescence intensity between venom and control treated primary hemocyte cultures.

All statistics were done in the R statistical computing environment [106] using the plyr [108], reshape2 [109], multcomp [110], ARTool [111], lme4 [112], lmerTest [113], and effects [114] packages. Graphs were produced using the ggplot2 and ggpubr packages [115,116].

## Phylogenetics

Phylogenetic analyses were conducted in MEGA X [117,118] using COI DNA sequences for the parasitoid strains used in the study (S1 Table). Evolutionary history was inferred by using the Maximum Likelihood method and Kimura 2-parameter model with 1000 bootstrap replicates [119]. The initial tree for the heuristic search was obtained automatically by applying Neighbor-Join and BioNJ algorithms to a matrix of pairwise distances estimated using the Maximum Composite Likelihood (MCL) approach in MEGA X, and then selecting the topology with superior log likelihood value. Branches corresponding to partitions reproduced in less than 50% of the bootstrap replicates were collapsed. All positions containing gaps and missing data were eliminated. The resulting phylogenetic trees were visualized using FigTree (version 1.4.3, http://tree.bio.ed.ac.uk/).

## Supporting information

**S1 Table. Species and strains names for all parasitoids used in this study.** Accession numbers for COI sequence data are given.
(DOCX)

**S1 Fig.** Brightfield (A-C) and fluorescence (A'-C') images of *10xSTAT92E-GFP* larvae. Strong fluorescence is seen in gut cells (A,A'). No fluorescence in seen in lymph gland dissected from naïve (B,B') or LcNet infected (C,C') larvae.
(TIF)

**S2 Fig.** Brightfield (A-C) and fluorescence (A'-C') images of fat bodies dissected from *10xSTAT92E-GFP* larvae. Images are representative of the expression categories used in Table 1. A' is representative of '-', B' is representative of '+' and C' is representative of '++'.
(TIF)

**S3 Fig. Fluorescent images of body wall muscle from *10xSTAT92E-GFP* larvae.** Images are representative of the expression categories used in Table 2. A is representative of '-', B is representative of '+' and C is representative of '++'. All images were taken at the posterior end. (TIF)

## Acknowledgments

We thank members of the Mortimer lab for comments on the manuscript, Drs. Alysia Vrailas-Mortimer and Erika Bach for *D. melanogaster* stocks, and Drs. Dan Hultmark, Jacques van Alphen and John Pool for parasitoid strains. We would like to thank the editor and two anonymous reviewers for their supportive comments and constructive suggestions on our work. We feel that our manuscript has been substantially improved by this review process. Stocks obtained from the Bloomington Drosophila Stock Center (NIH P40OD018537) were used in this study. The ISU Confocal Microscopy Facility was funded by NSF grant DBI-1828136.

## Author Contributions

**Conceptualization:** Susanna E. Brantley, Todd A. Schlenke, Nathan T. Mortimer.

**Data curation:** Nathan T. Mortimer.

**Formal analysis:** Susanna E. Brantley, Pooja Kr, Nathan T. Mortimer.

**Funding acquisition:** Todd A. Schlenke, Nathan T. Mortimer.

**Investigation:** Susanna E. Brantley, Corinne M. Stouthamer, Pooja Kr, Mary L. Fischer, Joshua Hill, Nathan T. Mortimer.

**Supervision:** Todd A. Schlenke, Nathan T. Mortimer.

**Visualization:** Susanna E. Brantley, Nathan T. Mortimer.

**Writing – original draft:** Susanna E. Brantley, Nathan T. Mortimer.

**Writing – review & editing:** Corinne M. Stouthamer, Pooja Kr, Mary L. Fischer, Joshua Hill, Todd A. Schlenke.

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
