## [Decision Letter · Decision Letter 0]

30 Jan 2024

Dear Mortimer,

Thank you very much for submitting your manuscript "Host JAK/Stat activity is a target of parasitoid wasp virulence strategies" for consideration at PLOS Pathogens. As with all papers reviewed by the journal, your manuscript was reviewed by members of the editorial board and by several independent reviewers. The reviewers appreciated the attention to an important topic. Based on the reviews, we are likely to accept this manuscript for publication, providing that you modify the manuscript according to the review recommendations.

Both reviewers were supportive of this manuscript. Reviewer two brings up a genetic point that I would like to see discussed, at least in rebuttal. Why are only gain-of-function alleles included in this study. Formally, I think this says that this signaling pathway can sufficient but doesn't address necessity. To me, it is the loss of function phenotype that suggests a process is naturally involved in the immune response. Can you at least address this problem and explain why you didn't test loss-of-function mutations?

Sincerely,

David S. Schneider

Academic Editor

PLOS Pathogens

James Collins III

Section Editor

PLOS Pathogens

Michael Malim

Editor-in-Chief

PLOS Pathogens

orcid.org/0000-0002-7699-2064

Both reviewers were supportive of this manuscript. Reviewer two brings up a genetic point that I would like to see discussed, at least in rebuttal. Why are only gain-of-function alleles included in this study. Formally, I think this says that this signaling pathway can sufficient but doesn't address necessity. To me, it is the loss of function phenotype that suggests a process is naturally involved in the immune response. Can you at least address this problem and explain why you didn't test loss-of-function mutations?

Reviewer Comments (if any, and for reference):

Reviewer's Responses to Questions

**Part I - Summary**

Reviewer #1: Encapsulation by hemocytes is a primary immune defense in insects against endoparasitoids. In Drosophila, lamellocytes are a hemocyte type that is required for capsule formation, while lamellocyte production is further known to be regulated by the JAK-STAT pathway. Several species and strains of larval endoparasitoids are known to parasitize D. melanogaster. In a previous study, the authors reported the Lh14 strain of Leptopilina heterotoma and Lb17 strain of Leptobilina boulardi disable by unknown mechanisms induction of unpaired (upd) which are secreted cytokines that activate the JAK-STAT pathway and lamellocyte production. This comparative study elaborates further on these findings by examining these and other species/strains of Drosophila endoparasitoids to assess how they affect JAK-STAT signaling. In the first part of the study, the authors used an ‘avirulent’ strain of Leptopilina clavipes named LcNet and Stat92E reporter flies to show that Stat92E expression in hemocytes and the larval body wall increase lamellocyte but not other hemocyte types in circulation (Fig. 1, 2). In the second part of the study the authors report variable effects on induction of JAK-STAT signaling in different tissues (hemocytes, fat body, body wall) after parasitism by different species/strains of wasps (Fig. 2-5).

My overall assessment is that the study brings new comparative results to the literature that further advance the field by showing that virulent species/strains of parasitoids evade encapsulation by differentially affecting the JAK-STAT pathway. The study does not identify the underlying virulence factors responsible for these effects. However, the results position the authors to identifying the virulence factors involved and their functions in future investigations. Approaches and interpretation of results were overall appropriate and support key conclusions in the opinion of this reviewer. The main areas for improvement are with the presentation. Key suggestions are itemized below.

Reviewer #2: The manuscript presents data to support that parasitoid wasps can inhibit the JAK-STAT pathway which plays a role in lamellocyte proliferation, which is the cellular immune response against parasitoid wasps in Drosophila melanogaster. The intro/background should probably include more on the different parasitoid wasps; in particular is D.melanogaster a natural host for them in the wild. Strengths include examination of different species and strains of parasitoid wasps and use of genetic tools (STAT GFP reporter, gain-of-function constructs or mutants for the pathway) to tease out the cell types and do epistasis. Their investigation of the interplay between parasitoid wasp and the host immune response and the specific role and targeting of the JAK-STAT pathway was pretty interesting. The main weakness was the reliance on one assay for much of the work (GFP reporter) and the use of only gain-of-function mutations. With all the different strains it did seem like more information could be presented, as the more functional assays (i.e., encapsulation) were limited to just a single parasitoid strain. The support of the manuscript’s main hypothesis (of parasitoid wasp inhibition of JAK-STAT as a way to counter the host immune response) would be stronger if they showed the data for more strains (along the avirulence to virulence spectrum) and still saw a clear correlation with many strains that inhibition of JAK-STAT resulted in decreased encapsulation. The work also did not present loss-of-function mutations and if that resulted in loss of an immune response to the avirulent strains. The experiments mentioned above seem rather obvious so it did make me wonder if the results of those experiments failed to support the manuscript’s main hypothesis or if they were being saved for another manuscript.

**Part II – Major Issues: Key Experiments Required for Acceptance**

Reviewer #1: Approaches and interpretation of results were overall appropriate and support key conclusions in the opinion of this reviewer.

Reviewer #2: They should look at loss-of-function of JAK-STAT pathway to determine if it's needed for the anti-parasitoid wasp immune response.

They should look at encapulation which would be more of an end functional assay for all the strains ideally (or at least strains that represent the different levels of inhibition of the JAK-STAT pathway) to see if their hypothesis of inhibition of JAK-STAT results in more parasitoid success in the host.

**Part III – Minor Issues: Editorial and Data Presentation Modifications**

Reviewer #1: The main areas for improvement are with the presentation. Key suggestions are itemized below.

Introduction

1. I thought the biggest weakness of the Introduction is that it doesn’t present a few key things readers need to fully follow what is presented in the Results. First, the authors present very little information about the different species and strains of larval endoparasitoids that parasitize D. melanogaster and associated differences in virulence. In turn, neither the Introduction nor Results provide the reader much information about the species/strains the authors used in the study but I conclude this kind of information is needed to understand why the authors chose to include the species/strains they used and why these choices were made. The authors also don’t mention their prior findings about Lh14 and Lb17 suppressing upd expression (reference 64) until the middle of the Discussion section, when instead it’s an obviously crucial part of the rationale for the experiments that were conducted in this study. It thus would be much more logical in my opinion to get key these points into the Introduction and in effect make clearer the goal of this study was to further advance prior findings by asking if all larval endoparasitoids that show evidence of suppressing the production/activity of lamellocytes do so by: 1) targeting JAK-STAT signaling and 2) targeting the same or different stages of JAK-STAT signaling. As written, the reader more or less has to put this rationale together on their own after reading the whole paper. Just citing references 12, 26, 38, 46 and 64-75 do not adequately provide the information readers need because these studies focus on only a subset of the species/strains the authors included in this manuscript. I thus suggest revising the Introduction to better introduce key features of the parasitoids the authors report results for and the prior findings they conducted on Lh14 and Lb17, so that the reader better understands overall rationale.

2. The authors also could do a better job of explaining what determines whether a given species/strain of wasp is ‘virulent’ versus ‘avirulent’. As a reviewer, I know this comes down to prior studies, mostly published, showing that species/strains which successfully develop and emerge at high frequencies (that is, successfully parasitize D. melanogaster) are virulent while those that don’t are avirulent. In many cases, but not all, avirulence has further been shown to result in a high frequency of wasp eggs being encapsulated. As alluded to by the authors in the Methods (pg 21) the primary hosts for the ‘avirulent’ species they used isn’t D. melanogaster. But some readers are not going to be well-versed on these points and what supports them in the greater literature. So, I think it’s incumbent on the authors to be clearer about this literature and the evidence supporting their classification of the species/strains they used in the study as virulent versus avirulent in the introduction, so that everyone can better follow the comparative studies the authors prioritized in the results shown in the second part of the study.

Results

3. The authors mostly present their results as means ± an error value but it’s increasingly more standard in the literature to present actual data points in graphs that give readers a much better sense of how robustly assays were replicated and spread in the data being presented. Some journals have further made this an absolute requirement for publication. I would thus suggest the authors need to do this for most/all of the data they’re presenting in the paper.

4. Given the importance of the wasp species and strains used in the study, I would elevate Fig. S2 to being a main figure, that readers would be able to more readily see when reading the main text.

5. I didn’t find Table 1 and Table 2 to be very helpful and would suggest presenting the actual fluorescence intensity data in Fig. S3 and S4 along with the qualitative results that are currently shown. This approach would also be complementary to to how the hemocyte data are presented in Fig. 3A. More complete S3 and S4 figures and dropping Table 1 and 2 would in my opinion be better for readers.

6. I thought it was an odd choice to present total hemocyte counts and lamellocyte numbers in Fig. 2 and 4 as cells per ml when fly larvae are so small and numbers of circulating hemocytes are on levels of ul of hemolymph (or less) per larva. Just seems slghtly misleading to present the data on a per ml basis when the methods suggest hemolymph volumes weren’t even measured (triplicate pools of 5 larvae bled into PBS). I would suggest hemocytes per larva would be more appropriate for what the authors actually counted (or a better explanation of how cells per ml were estimated given the methods reported on page 22). The authors also need to clearly state in the methods how numbers of lamellocytes were determined and distinguished from other cell types given total cell counts are reported to derive from using a hemocytometer where distinguishing cell types solely on the basis of morphology can be difficult.

7. Total number of larvae dissected per replicate and more detail is needed in the methods for readers to fully follow the encapsulation data reported in Fig. 4D.

8. Given the similar effect of GxUG, GxHaw LgCam and LgSA on hopTum flies (Fig. 4), I wasn’t clear why the venom assays the authors report were restricted to only LgCam. Would seem at minimum the authors want to look at and report outcomes using venom from at least one of the Ganapsis xanthopoda strains as well to make sure venom from this species also contains a component that similarly (or not) inhibited JAK-STAT as reported for LgCam in Fig. 4C.

Discussion

Overall, pretty clear but as noted above certain points not mentioned until the discussion like earlier findings for Lh14 and Lb17 should in my opinion be moved to the introduction. The discussion linking the assays using avirulent wasps to the greater literature on lamellocyte production/origins was fine in my opinion and does a good job of synthesizing key points/conclusions outside of maybe needing to bring in the point there are 3 unpaired paralogs with most evidence to my knowledge focusing on upd2 and 3 in regard to immune functions. This maybe should get worked into what the authors have written and potential routes of origin for lamellocytes from cells in hematopoietic organs versus sessile hemocytes elsewhere in the body. Other small points the authors potentially should mention in regard to successful parasitism is that the author’s results focus on the idea individual wasp species/strains employ very specific virulence strategies against D. melanogaster. I’m sure this is partially true but the literature also indicates D. melanogaster and other drosophilids exhibit genetic differences across their geographic ranges that also considerably influence permissiveness to. Thus, virulence and avirulence as used in this study is affected in part by the authors conducting their experiments in Canton strain flies. It also means that while

---

## [Decision Letter · Decision Letter 1]

17 Jun 2024

Dear Mortimer,

We are pleased to inform you that your manuscript 'Host JAK/Stat activity is a target of parasitoid wasp virulence strategies' has been provisionally accepted for publication in PLOS Pathogens.

Best regards,

James J Collins III

Section Editor

PLOS Pathogens

James Collins III

Section Editor

PLOS Pathogens

Michael Malim

Editor-in-Chief

PLOS Pathogens

orcid.org/0000-0002-7699-2064

Reviewer Comments (if any, and for reference):

Reviewer's Responses to Questions

**Part I - Summary**

Reviewer #1: I had no major concerns about the original submission but listed several minor concerns that primarily focused on presentation, some of the figures, data analysis, and the opinion that vFig. 4 should extend beyond LgCam. The authors adopted many but not all of my original suggestions. The revisions improve the manuscript and I'm fine with the authors not adopting everything I originally suggested as the points that weren't adopted are not critical to the primary results the authors present or overall interpretation. I also was okay with the authors' response to the other reviewer's points about the lack of loss of function data in this study, as I was aware of other studies in the literature that had examined the effects of loss of function JAK-Stat mutants on encapsulation in melanogaster that the authors now cite. I thus am overall satisfied with the revised manuscript.

Reviewer #2: I'm happy with their revisions to address my previous concerns.

**Part II – Major Issues: Key Experiments Required for Acceptance**

Reviewer #1: See above

Reviewer #2: (No Response)

**Part III – Minor Issues: Editorial and Data Presentation Modifications**

Reviewer #1: See above

Reviewer #2: (No Response)

PLOS authors have the option to publish the peer review history of their article (what does this mean?). If published, this will include your full peer review and any attached files.

Reviewer #1: No

Reviewer #2: No

---

## [Editor Report · Acceptance letter]

26 Jun 2024

Dear Mortimer,

We are delighted to inform you that your manuscript, "Host JAK-STAT activity is a target of parasitoid wasp virulence strategies," has been formally accepted for publication in PLOS Pathogens.

Best regards,

Michael Malim

Editor-in-Chief

PLOS Pathogens

orcid.org/0000-0002-7699-2064